# Antischistosomal, antionchocercal and antitrypanosomal potentials of some Ghanaian traditional medicines and their constituents

**Emmanuella Bema Twumasi[1], Pearl Ihuoma Akazue[2], Kwaku Kyeremeh[1], Theresa Manful Gwira[2], Jennifer Keiser[3], Fidelis Cho-Ngwa[4], Adrian Flint[5], Barbara Anibea[1], Emmanuel Yeboah Bonsu[1], Richard K. Amewu[1], Linda Eva Amoah[6], Regina Appiah-Opong[6], Dorcas Osei-Safo[1] ***

**1** Department of Chemistry, University of Ghana, Accra, Ghana, **2** West African Centre for Cell Biology and Infectious Diseases, Department of Biochemistry, Cell and Molecular Biology, University of Ghana, Accra, Ghana, **3** Helminth Drug Development Unit, Department of Medical Parasitology and Infection Biology, Swiss Tropical and Public Health Institute, Basel, Switzerland, **4** Biotechnology Unit laboratories, Faculty of Science, University of Buea, Buea, Cameroon, **5** School of Sociology, Politics and International Studies, University of Bristol, Bristol, United Kingdom, **6** Noguchi Memorial Institute for Medical Research, University of Ghana, Accra, Ghana

* dosei-safo@ug.edu.gh

## Abstract

### Background

Ghana is endemic for some neglected tropical diseases (NTDs) including schistosomiasis, onchocerciasis and lymphatic filariasis. The major intervention for these diseases is mass drug administration of a few repeatedly recycled drugs which is a cause for major concern due to reduced efficacy of the drugs and the emergence of drug resistance. Evidently, new treatments are needed urgently. Medicinal plants, on the other hand, have a reputable history as important sources of potent therapeutic agents in the treatment of various diseases among African populations, Ghana inclusively, and provide very useful starting points for the discovery of much-needed new or alternative drugs.

### Methodology/Principal findings

In this study, extracts of fifteen traditional medicines used for treating various NTDs in local communities were screened *in vitro* for efficacy against schistosomiasis, onchocerciasis and African trypanosomiasis. Two extracts, NTD-B4-DCM and NTD-B7-DCM, prepared from traditional medicines used to treat schistosomiasis, displayed the highest activity ($IC_{50}$ = 30.5 µg/mL and 30.8 µg/mL, respectively) against *Schistosoma mansoni* adult worms. NTD-B2-DCM, also obtained from an antischistosomal remedy, was the most active against female and male adult *Onchocera ochengi* worms ($IC_{50}$ = 76.2 µg/mL and 76.7 µg/mL, respectively). Antitrypanosomal assay of the extracts against *Trypanosoma brucei brucei* gave the most promising results ($IC_{50}$ = 5.63 µg/mL to 18.71 µg/mL). Incidentally, NTD-B4-DCM and NTD-B2-DCM, also exhibited the greatest antitrypanosomal activities ($IC_{50}$ =

**Funding:** DOS, KK, RKA, AF, LEA, RAO are grant recipients of the Worldwide Universities Network Research Development Fund 2017 from the Worldwide Universities Network (UK) and grant number 18-191 RG/CHE/AF/AC_G - FR 3240303659 from The World Academy of Sciences. The funders had no role in study design, data collection and analysis, decision to publish, or preparation of the manuscript.

**Competing interests:** The authors have declared that no competing interests exist.

5.63 μg/mL and 7.12 μg/mL, respectively). Following the favourable outcome of the antitrypanosomal screening, this assay was selected for bioactivity-guided fractionation. NTD-B4-DCM, the most active extract, was fractionated and subsequent isolation of bioactive constituents led to an eupatoriochromene-rich oil (42.6%) which was 1.3-fold ($IC_{50}$ <0.0977 μg/mL) more active than the standard antitrypanosomal drug, diminazene aceturate ($IC_{50}$ = 0.13 μg/mL).

## Conclusion/Significance

These findings justify the use of traditional medicines and demonstrate their prospects towards NTDs drug discovery.

## Author summary

The discovery of new drugs is vital to achieving the World Health Organization eradication targets for neglected tropical diseases (NTDs). Traditional medicines serve as sources of primary healthcare needs for most affected populations and therefore represent a valuable resource in this regard, albeit largely underdeveloped. We investigated extracts of a selection of traditional medicines for efficacy and safety to justify their use for treating NTDs in Ghana. We screened the extracts for activity against schistosomiasis (*Schistosoma mansoni*), onchocerciasis (*Onchocera ochengi*) and trypanosomiasis (*Trypanosoma brucei brucei*). Generally, more extracts effectively inhibited *O. ochengi* than *S. mansoni* lifecycle stages. Further, *S. mansoni* juvenile and *O. ochengi* adult-stage male worms were more susceptible to the extracts than adult-stage *S. mansoni* and *O. ochengi* female worms. Overall, the extracts were most active against the trypanosomes. Hence, we selected the antitrypanosomal assay to identify active principles and this resulted in a non-toxic oil which was more active than the extract, and also more effective than the standard antitrypanosomal drug, diminazene aceturate. The major oil constituent, eupatriochromene, has demonstrated antitrypanosomal activity in other studies. We recommend that the quality of traditional medicines is improved.

## Introduction

Endemic neglected tropical diseases (NTDs) in Ghana include schistosomiasis (SCH), onchocerciasis (ONCHO) and lymphatic filariasis (LF) [1–7]. SCH is caused by the blood flukes *Schistosoma haematobium* which is responsible for the highly prevalent urinary schistosomiasis, and *S. mansoni*, which causes intestinal schistosomiasis. In both conditions, chronic infections are fatal in severe cases [1,8]. ONCHO, commonly referred to as river blindness, is caused by the parasitic worm *Onchocerca volvulus*. The disease results in severe disability characterized by intense itching, disfiguring skin conditions and visual impairment that can lead to permanent blindness [9]. LF, also known as elephantiasis, is caused by the parasitic filarial worm *Wuchereria bancrofti*. Symptoms include impairment of the lymphatic system which manifests as abnormal enlargement of body parts accompanied by severe disability and pain. Patients also suffer immense psychological and social stigma [10].

Mapping of these diseases in Ghana over the past decade indicated nationwide prevalence (in all 170 districts at the time) for SCH with an endemic population of 7 million [1]. LF was identified to be endemic in 74 out of the 170 districts, affecting about 12.3 million people while

the population at risk with ONCHO was 8.2 million in 73 districts [1]. Intervention pro-grammes for the NTDs are integrated into the Ghana Health Service strategic plan with a goal 'to prevent, control, eliminate or eradicate the Neglected Tropical Diseases from Ghana by the year 2020' in consonance with the World Health Organization (WHO) roadmap [1]. Mass drug administration (MDA) is the WHO-recommended preventive chemotherapy (PC) strat-egy to stop transmission of SCH, ONCHO and LF. MDA involves treatment of all at risk-pop-ulations in endemic communities with praziquantel for SCH, ivermectin and albendazole for LF and ivermectin for ONCHO [11]. In 2010, 11.9 million people required PC for LF in Ghana. By 2018, this number had dropped to 1.4 million, a massive reduction of 88.2% in the population needing PC for LF [12]. On the other hand, in the same period, PC requirement for ONCHO cases shot up by 81.5% from 1.5 million to 8.1 million [13] while the population at-risk for SCH increased by 39.0% from 6.4 million to 10.5 million [14]. The percentage PC coverage in 2010 recorded 62.8% for LF, 100% for ONCHO and 27.4% for SCH. PC coverage statistics for 2018 are not yet available. Regardless of the encouraging decrease in disease bur-den for LF in 2018, Ghana remained endemic for the disease in 2019, together with SCH and ONCHO [15]. The Ghana national launch of the 2019 MDA programme revealed that the country 'will not be able to eradicate NTDs by 2020 as required by the World Health Organiza-tion' [16]. This is because more cases of LF are being detected and ONCHO is still endemic in many districts irrespective of conducting MDA annually [16]. Recent surveys in endemic com-munities in Ghana have suggested reduced efficacy for ivermectin against *O. volvulus* while persistent LF 'hotspots' remain after almost two decades of MDA [17–19]. Moreover, SCH remains endemic [20] and praziquantel does not prevent reinfection, infection rates tending to return to high values within 24 months [2]. Identified threats to the MDA programme include negative propaganda about the safety of the drugs, experiences of side reactions associated with treatment, lack of donor drugs for some NTDs and inability of the government to sustain the programme after cessation of donor support [11].

One of the key aspects regarding the combatting of NTDs is the *neglected* aspect. Although these diseases plague millions across low- and middle-income countries, they are of little inter-est to the major drug companies. The drugs that are commonly used in the MDA to fight NTDs in Ghana and other endemic countries were all patented in the 1970s and 1980s: albendazole was patented in 1975, ivermectin in 1982, and praziquantel in 1973. In part, this neglect can be attributed to the high cost of bringing new drugs to the market which has been estimated to be as high as $2.87 billion [21]. This lack of progress is deeply problematic because, as outlined at the WHO summit on NTDs in 2012 which resulted in the 'London Declaration on Neglected Tropical Diseases', new drugs are needed if these diseases are to be eradicated [22]. Given fund-ing constraints, it is clear that innovative approaches to the development of new compounds are required if the problem posed by NTDs is to be comprehensively addressed.

Ghana abounds in indigenous medicinal plant species whose products are utilized by a vast majority of the population to meet their healthcare needs due to factors such as easy access, affordability, and perceived safety and efficacy over orthodox drugs [23–25]. The WHO recog-nizes the importance of medicinal plants and encourages developing countries to integrate into their mainstream healthcare systems traditional medicines (TMs) that have established safety and efficacy profiles [26,27]. In 2010, Ghana embraced this model of supplementing conventional medicines with TMs and is promoting their coexistence; hence some TM prod-ucts are currently included in Ghana's essential drug list [28,29]. Such TMs, however, are those used in treating priority diseases and common ailments such as malaria, hypertension and diarrhoea, and are certified and approved for use by the Ghana Food and Drugs Authority (FDA). Of particular mention is *Cryptolepis sanguinolenta*, which has been demonstrated to be clinically efficacious against malaria and marketed either as a decoction prepared from the

aqueous root extract, 'Mist Nibima' [30,31] or the tea formulation trademarked as Phyto-Laria [32]. On the contrary, NTD-related TMs do not enjoy this level of recognition by the FDA, largely due to lack of requisite efficacy testing platforms for NTDs. Regardless, the remedies receive high patronage by populations affected by NTDs, efficacy claims backed mainly by anecdotal evidence. Consequently, engaging with TM practitioners represents an alternative way of circumventing the usual templates for identifying new chemical entities.

TMs that are locally used against NTDs, contain compounds that possess activity against the disease-causing pathogens; hence, this study was undertaken to investigate the efficacy of a selection of TMs used in Ghana for the treatment of the endemic NTDs—SCH, ONCHO and LF. Testing of the TMs was further extended to African trypanosomiasis. African trypanosomiasis is a disease caused by certain species of the protozoan parasite called trypanosomes and there are two forms of African trypanosomiasis: the human form of the disease known as Human African Trypanosomiasis (HAT) and the animal form of the disease (AAT). Worldwide, there has been a significant decrease in the number of reported cases of HAT [33] and in Ghana, the disease is considered to be near elimination as reports indicate that no incidence has been recorded since 2014 [34]. Despite this apparent progress in the control of HAT, there is a risk of reoccurrence of endemicity as was observed two decades ago. On the other hand, AAT is currently recording increasing prevalence rates globally–Ghana inclusively [35,36]. AAT has a huge negative impact on food security and socioeconomic development in endemic countries. Also, there are concerns about AAT-infected animals serving as reservoirs for species of trypanosomes that cause human infections [37]. As with other NTDs, there are no effective drugs for the management of this disease, hence new treatments are urgently sought to curtail African trypanosomiasis.

The findings from this study justify the exploration of TMs to identify potential medications against NTDs.

## Methods

### Ethical approval

Ethical approval for the study was provided by the Ethics Committee for Basic and Applied Sciences, University of Ghana, with reference number ECBAS 045/17-18. Formal consent was obtained verbally.

### Traditional medicines

The TMs investigated in the study were obtained from members of the Ghana Federation of Traditional Medicines Practitioners Association (GHAFTRAM), following an interaction and administration of a questionnaire. A total of 15 TMs used for the treatment of schistosomiasis (7), onchocerciasis (6) and LF (2) were collected (Table 1). Formulations comprised between one to five plant species and were available in two forms: aqueous herbal preparations and dried powdered herbs. NTD-O3 and NTD-O4 were both prepared from the leaves of *Delonix regia* by different practitioners for use against onchocerciasis. The latter was the dry leaves while NTD-O3 was the aqueous extract. According to the practitioners, their efficacy claims are based on the number of people who experience reduction or relief from symptoms as a result of using their products.

### Extraction procedure

About 100 g of each powdered dried herb was extracted successively with 500 mL dichloromethane (DCM) and methanol (MeOH) *via* maceration for 72 hours. The extracts were

**Table 1. List of traditional medicines.**

| No. | Code | Plant species | Nature | Dosage |
|---|---|---|---|---|
| **1. Bilharzia/Schistosomiasis–NTD-B series** | | | | |
| 1 | NTD-B1 | *Alchornea cordilfolia, Momordica charantia* | Aqueous | 4 tbsp x 3 x 14 days |
| 2 | NTD-B2 | *Syzygium aromaticum, Xylopia aethiopica, Tapinanthus bangwensis, Phyllanthus niruri* | Aqueous | 3 tbsp x 3 |
| 3 | NTD-B3 | *Trichila monadelpha, Alstonia boonei, Picralima nitida* | Dried herbs | 1 tbsp in 250 mL of water x 2 |
| 4 | NTD-B4 | *Aloe vera, Taraxacum officinale* | Dried herbs | 1 tsp in 100 mL of water x 3 |
| 5 | NTD-B5 | *Combretum* sp, *Smeathmannia* sp, *Morinda lucida, Mitragyna stipulosa, Paulina pinnata* | Aqueous | 4 tbsp x 4 |
| 6 | NTD-B6 | *Anthocleista nobilis, Nauclea latifolia, Rauwolfia vomitoria, Alstonia boonei, Ageratum conyzoides* | Aqueous | 4 tbsp x 3 |
| 7 | NTD-B7 | *Vernonia amygdalina, Khaya senegalensis, Mangifera indica, Azadirachta indica* | Aqueous | 5 tbsp x 3 |
| **2. Onchocerciasis/River blindness—NTD-O series** | | | | |
| 8 | NTD-O1 | *Mangifera indica, Momordica charantia, Zingiber officinale, Xylopia aethiopica* | Aqueous | 2 tbsp x 3 |
| 9 | NTD-O2 | *Bambusa vulgaris, Xylopia aethiopica, Citrus aurantifolia* | Aqueous | 5 tbsp x 3 after meals |
| 10 | NTD-O3 | *Delonix regia* | Aqueous | 6 tbsp x 3 |
| 11 | NTD-O4 | *Delonix regia* | Dried herbs | 1 tbsp in 200 mL of hot water x 2 |
| 12 | NTD-O5 | *Uvaria chamae, Momordica charantia* | Aqueous | 4 tbsp x 4 after meals |
| 13 | NTD-O6 | *Vernonia amygdalina, Khaya senegalensis, Mangifera indica, Azadirachta indica* | Aqueous | 5 tbsp x 3 |
| **3. Elephantiasis/Lymphatic filariasis—NTD-E series** | | | | |
| 14 | NTD-E1 | *Spathodea campanulata* | Aqueous | 4 tbsp x 3 after meals |
| 15 | NTD-E2 | *Newbouldia leavis* | Dried Herbs | Mix 2 tbsp with water into a paste & rub on affected area. Then take 1 tbsp and chew |

* tbsp- tablespoon *tsp- teaspoon

filtered through a Whatman No. 1 filter paper and fresh solvent added at 24-hour intervals. For the aqueous herbal preparations, about 500 mL of each sample was partitioned three times with 500 mL of DCM, followed by the same volume of n-butanol (n-BuOH). The extracts were also filtered and all filtrates were concentrated to dryness *in vacuo*, refrigerated at 4˚C and used within one month of preparation. All solvents used for the extraction processes were of HPLC grade.

## Biological activity screening of the crude extracts

Platforms for screening for activity against SCH, ONCH and African trypanosomiasis were identified. Attempts to identify a laboratory for antiLF assay proved futile hence, no biological activity assay against LF was undertaken. Antischistosomal potential of the TMs was evaluated by screening them *in vitro* against newly transformed schistosomula (NTS) and adult worms of *S. mansoni* at the Swiss Tropical and Public Health Institute, Switzerland. Filaricidal activities and cytotoxicity of the TMs on *Onchocerca ochengi* were assayed at the Biotechnology Unit, Faculty of Science, University of Buea, Cameroon. The TMs were further evaluated for their *in vitro* antitrypanosomal activity against bloodstream forms of *Trypanosoma brucei brucei* at the West African Centre for Cell Biology of Infectious Pathogens (WACCBIP), University of Ghana. All the crude extracts prepared from the TMs were subjected to the three biological activity screening platforms.

### *In vitro* antischistosomal activity screening

**Sample preparation.** The crude extracts were dissolved in dimethyl sulfoxide (DMSO). Extracts were diluted 1:10 in the schistosome media used as described below. Final

concentration of DMSO in the assay did not exceed 1%. Extracts were tested on NTS and adult worms of *S. mansoni* to evaluate their antischistosomal activity. Studies were carried out in accordance with Swiss national and cantonal regulations (permission number 2070) on animal welfare. Praziquantel and DMSO served as positive and negative controls, respectively.

**Schistosome cultures.** The NTS culture medium was prepared by supplementing M199 medium (Medium 199 Earles and Hepes) with 5% fetal calf serum (iFCS, 100 U/ml), 1% (v/v) streptomycin/penicillin mixture (Invitrogen, 100 U/ml) and 1% Mäser Mix. Adult worm culture medium was prepared by supplementing Roswell Park Memorial Institute (RPMI) 1640 medium with 5% (v/v) fetal calf serum and 1% (v/v) streptomycin/ penicillin. Both media cultures were maintained at 4˚C.

**Antischistosomal screening against NTS of *S. mansoni*.** The protocol employed in the evaluation of the antischistosomal activity of the crude extracts has been described by Lombardo et al [38]. About 175 µL of the supplemented M199 medium was added to each well of the 96-well plate followed by the addition of 25 µL of the prepared extract stock solution, and 30–40 NTS per well. DMSO controls were prepared by the addition of 25 µL of (10% v/v DMSO and 90% medium) in 175 µL of supplemented M199 medium. Two biological replicates were carried out and incubated at 37˚C under 5% carbon dioxide ($CO_2$). They were then evaluated at 72 hours after drug incubation. The parasites were evaluated by means of a brightfield microscope with x 4 or x 10 magnification. Scores were assigned to each well to reflect the phenotype of most of the parasites in the well and compared to the controls as follows: 3 = motile, no changes to morphology or transparency; 2 = reduced motility and/or some damage to tegument, as well as reduced transparency and granularity; 1 = severe reduction of motility and/or damage to tegument, with high opacity and high granularity; 0 = dead [38].

**Antischistosomal screening against adult worms of *S. mansoni*.** About 20 µL of the stock solution (10 mg/mL) of the crude extracts were pipetted into 24-well plates and 1980 µL of the supplemented RPMI 1640 was added. All the crude extracts were tested in triplicates and evaluated 72 hours later. In the control experiment, 10% v/v of DMSO in the supplemented RPMI 1640 medium was prepared and 15 µL of this was pipetted into the control wells to achieve 0.1% concentration. Three pairs of worms (both male and female) were carefully placed in each well and incubated at 37˚C under 5% $CO_2$. The sexes of both worms were noted, and the following phenotypic scores assigned at 72 hours: 0 = dead, the worms appear darkened and motility of the ventral and oral sucker is absent; 0.25–1 = reduced motility and significant tegument damage on different severity level; 1.25–2 = reduced motility or marked tegument damage on different severity level; 2.25–3 = viable, nice tegument, good motility, no big changes to morphology, transparency and intact tegument, active ventral and oral sucker; 3 = motile, no changes to morphology, transparency and intact tegument, active ventral and oral sucker. Samples were considered hits if an activity of at least 70% was observed [38].

### *In vitro* antionchocercal activity screening

**Sample preparation.** Stock solutions of 25 mg/mL were prepared from each crude extract in DMSO and tested on worms as well as the larvae. The positive control which was used in this assay was 10 µM auranofin while 2% DMSO was employed as negative control.

***Onchocerca* cultures.** The *O. ochengi* adult worms were recovered from infected cattle skin by dissection and submerged in complete culture medium (CCM), consisting of RPMI 1640 with $NaHCO_3$ and supplemented with 25 mM HEPES, 0.3 g/L γ-irradiated L-glutamine powder, 5% newborn calf serum, 200 units/mL penicillin, 200 µg/mL streptomycin and 0.25 µg/mL amphotericin B; pH 7.4, in 24-well culture plates. These were incubated in humidified air at 37˚C under an atmosphere of 5% $CO_2$. The infected cattle skin was washed, sterilized

with ethanol, and then incubated for about 4 to 6 hours in CCM at room temperature as previously described [39]. Highly motile *O. ochengi* microfilariae which emerged were concentrated by centrifugation and re-suspended in CCM. They were then distributed into 96-well plates containing Monkey Kidney Epithelial cell (LLC-MK2) layer, and their viability and sterility ascertained for 24 h prior to addition of extracts.

**Primary screening against adult worms and microfilariae.** Primary screens were conducted to eliminate inactive extracts. Adult worms were treated in quadruplicates with the extracts at 200 μg/mL in CCM, 10 μM auranofin or 2% DMSO and incubated at 37°C, in an atmosphere of 5% $CO_2$ for 5 days. Microfilariae screens were conducted in duplicates. Activity scores of the extracts on the male adult worms were based on the motility of the worms observed under a microscope. The viability of the adult female worms was assessed biochemically by visual estimation of the percentage inhibition of formazan following the incubation of the nodules in 500 μL of 0.5 mg/mL MTT [39]. A 100% kill in the adult female worms was confirmed by the complete disappearance of a blue coloration since the worm appears blue in the negative control. An extract was considered active if there was ≥ 90% inhibition of male worm motility or of formazan formation; moderately active if there was a 50–89% inhibition of male worm motility or of formazan formation and inactive if there was a < 50% inhibition of male worm motility or of formazan formation.

**Secondary screening against adult worms.** Extracts which displayed 100% and >60% activity on adult male and female worms, respectively in primary screens were further tested at serial dilutions of eight concentrations (200, 100, 50, 25, 12.5, 6.25 and 3.125 μg/mL) in order to determine the $IC_{50}$ values.

**Cytotoxicity assessment of extracts.** Preliminary cytotoxicity studies of the crude extracts at 200 μg/mL were evaluated on LLC-MK2 cells.

## *In vitro* antitrypanosomal screening

**Sample preparation.** The crude extracts and fractions were dissolved in 100% dimethyl sulfoxide (DMSO) to a concentration of 20 mg/mL and stored in aliquots at -20°C; this constituted the parent stock. Working solutions of the extracts were prepared from the parent stock by dissolving in autoclaved distilled water to a concentration of 2 mg/mL.

## Trypanosome cultures

Wild-type bloodstream forms trypanosomes (*Trypanosoma brucei brucei*; GuTat 3.1 strain) were cultured in Hirumi's Modified Iscove's Medium-9, HMI-9 [40] supplemented with 1% penicillin-streptomycin and 10% heat-inactivated fetal bovine serum (Gibco). The trypanosome cultures were maintained at 37°C and 5% $CO_2$.

## Antitrypanosomal screening against *T. b. brucei*

Antitrypanosomal screening of the crude extracts, and later of the fractions of the TMs, was conducted using the Alamar blue assay [41] with slight modifications. Serial dilutions of the extracts were prepared on a 96-well plate with a starting concentration of 100 μg/mL to 0.1953 μg/mL for all samples, except for the positive control which had a starting concentration of 25 μg/mL to 0.0488 μg/mL. The final concentration of DMSO in the well with the highest percentage of DMSO was 0.5%. The cell density of trypanosomes in their logarithmic growth phase was adjusted to 2 x $10^5$ cells/ml and 100 μL of this suspension was added into each well, excluding the media control well to which no cell was added. The negative control consisted of only trypanosome cells in media with no treatment (extracts, fractions, or drug) added. Diminazene aceturate (DA), a standard antitrypanosomal drug, served as the positive

control. After 24 hours incubation, 20 μL of 500 μM Alamar blue dye was added and incubated for an additional 24 hours. Fluorescence readings were taken using a Varioskan multimode plate reader (ThermoFisher Scientific, USA) at an excitation wavelength of 530 nm and emission wavelength of 590 nm. The experiment was carried out in three biological replicates, with each biological replicate containing three technical replicates.

### Statistical analysis for antischistosomal, antionchoceral and antitrypanosomal screening

Antischistosomal percentage inhibition of the crude extracts on the test organisms were evaluated using the formula below.

$$\% \text{ Effect} = \frac{100 - \frac{\text{Average (test)} \times 100}{\text{Average control}}}{100}$$

The filaricidal activity data obtained were analysed using GraphPad Prism 6.0 (GraphPad Prism INC., CA, USA) to determine $IC_{50}$ values of active extracts. Plate readouts for the Alamar blue assay for antitrypanosomal screening were analyzed using non-linear regression analysis for growth inhibition on GraphPad Prism version 8.0. The antitrypanosomal activity was expressed as the $IC_{50}$ value of each extract and fraction and was determined for three biological replicates, each with triplicate determinations.

### Bioassay-guided fractionation and chemical profiling of active extract

About 15 g of the active crude extract was dissolved in a minimum volume of DCM and mixed with about 70 g of silica gel 60 (*Sigma-Aldrich)*. The slurry was air-dried, packed into cartridges and flashed sequentially with 375 mL each of 100% petroleum ether (PE), PE and ethyl acetate (EtOAc) mixtures (9:1, 8:2, 1:1 and 2:8), 100% EtOAc and EtOAc/MeOH 1:1. The resulting fractions were screened for activity. Fractions that exhibited biological activity were selected for chemical profiling to isolate and characterize their chemical constituents. Isolation was achieved by subjecting the fractions to silica gel column chromatographic separation while characterization of the isolated compounds was done by Gas Chromatography-Mass Spectrometry (GC/MS), Nuclear Magnetic Resonance (NMR) and High-Resolution Electrospray Ionization Mass Spectroscopy (HRESIMS). The compounds were further tested for activity to identify the bioactive ingredients.

GC/MS analysis of the oils were conducted on Shimadzu GC/MS-QP2010 Ultra fitted with a ZB5 column (60 m × 0.25 mm ID × 0.25 μm). Helium was employed as the carrier gas and the temperature ramp was 2˚C/min up to 260˚C. Samples were prepared as 5% w/v solution with DCM. Identification of components was done based on the Sat Set library employing linear retention index and mass spectral matching (Aromatic Plant Research Center, Utah, USA). 1D and 2D NMR spectral data were acquired in $CDCl_3$ on a 500 MHz Brüker spectrometer with reference to tetramethylsilane (Department of Chemistry, University of Ghana). Mass spectrometric data were acquired on a Waters Synapt G2 QTOF Spectrometer by electrospray ionization at a cone voltage of 15 V (Central Analytical Facilities of Stellenbosch University, South Africa.).

## Results

### Preparation of crude extracts from the traditional medicines

From the 15 TMs collected, 30 crude extracts were prepared using DCM and either MeOH or BuOH depending on the nature of the remedy as described in the experimental section and in

**Table 2. List of traditional medicines, resulting crude extracts and their percentage yields.**

| No. | Code | Nature | Extracts | Percentage yield (%) |
|---|---|---|---|---|
| | | **1. Bilharzia/Schistosomiasis–NTD-B series** | | |
| 1 | NTD-B1 | Aqueous | NTD-B1-DCM<br>NTD-B1-WB | 0.38<br>0.53 |
| 2 | NTD-B2 | Aqueous | NTD-B2-DCM<br>NTD-B2-WB | 0.12<br>0.34 |
| 3 | NTD-B3 | Dried herbs | NTD-B3-DCM<br>NTD-B3-MeOH | 1.85<br>1.92 |
| 4 | NTD-B4 | Dried herbs | NTD-B4-DCM<br>NTD-B4-MeOH | 5.63<br>3.73 |
| 5 | NTD-B5 | Aqueous | NTD-B5-DCM<br>NTD-B5-WB | 0.87<br>0.98 |
| 6 | NTD-B6 | Aqueous | NTD-B6-DCM<br>NTD-B6-WB | 1.72<br>0.46 |
| 7 | NTD-B7 | Aqueous | NTD-B7-DCM<br>NTD-B7-WB | 1.87<br>0.92 |
| | | **2. Onchocerciasis/River blindness—NTD-O series** | | |
| 8 | NTD-O1 | Aqueous | NTD-O1-DCM<br>NTD-O1-WB | 0.72<br>1.46 |
| 9 | NTD-O2 | Aqueous | NTD-O2-DCM<br>NTD-O2-WB | 0.38<br>0.72 |
| 10 | NTD-O3 | Aqueous | NTD-O3-DCM<br>NTD-O3-WB | 1.57<br>1.46 |
| 11 | NTD-O4 | Dried herbs | NTD-O4-DCM<br>NTD-O4-MeOH | 3.09<br>0.29 |
| 12 | NTD-O5 | Aqueous | NTD-O5-DCM<br>NTD-O5-WB | 1.23<br>1.63 |
| 13 | NTD-O6 | Aqueous | NTD-O6-DCM<br>NTD-O6-WB | 1.72<br>0.46 |
| | | **3. Elephantiasis/Lymphatic filariasis—NTD-E series** | | |
| 14 | NTD-E1 | Aqueous | NTD-E1-DCM<br>NTD-E1-WB | 1.28<br>0.64 |
| 15 | NTD-E2 | Dried Herbs | NTD-E2-DCM<br>NTD-E2-MeOH | 0.54<br>2.11 |

Table 2. All the crude extracts were separately evaluated for their antischistosomal, antionchocercal and antitrypanosomal activities.

## Antischistosomal activity of extracts

At a concentration of 100 μg/mL, 8 extracts displayed >70% inhibition of the motility of NTS, making them eligible for testing on adult worms *S. mansoni* (Table 3). In the follow-up test, 2 out of the 8 extracts, NTD-B4-DCM and NTD-B7-DCM, maintained this effect at 78.4% and 84.3% inhibition, respectively with corresponding $IC_{50}$ values of 30.5 μg/mL and 30.8 μg/mL. Under the experimental conditions, praziquantel displayed an $IC_{50}$ value of 2.2 and 0.1 μM [42], respectively against NTS and adult worms of *S. mansoni*.

## Antionchocercal activity of the extracts

All the crude extracts were tested in primary screens on microfilariae and adult worms of *O. ochengi* at concentrations of 200 μg/mL. Also tested was NTD-B7-DCM/S1 which precipitated from NTD-B7-DCM as white crystals and identified by NMR as benzoic acid. According to

**Table 3. Antischistosomal activity of crude extracts against *S. mansoni*.**

| Crude Extracts | Effect in % Conc. 100 µg/mL | | |
| --- | --- | --- | --- |
| | NTS | Adult | *IC$_{50}$ (µg/mL) |
| NTD-B1-DCM | 62.50 ± 7.6 | | |
| NTD-B1-WB | 53.57 ± 5.1 | | |
| NTD-B2-DCM | 32.14 ± 5.1 | | |
| NTD-B2-WB | 30.36 ± 7.6 | | |
| NTD-B3-DCM | 57.14 ± 0.0 | | |
| NTD-B3-MeOH | **89.59 ± 2.9** | 60.07 ± 0.0 | |
| **NTD-B4-DCM** | **100.00 ± 0.0** | **78.40 ± 2.8** | **30.5** |
| NTD-B4-MeOH | 35.72 ± 15.2 | | |
| NTD-B5-DCM | 32.15 ± 15.2 | | |
| NTD-B5-WB | 55.36 ± 2.5 | | |
| NTD-B6-DCM | 25.00 ± 0.0 | | |
| NTD-B6-WB | 20.84 ± 5.9 | | |
| **NTD-B7-DCM** | **100.00 ± 0.0** | **84.30 ± 0.0** | **30.8** |
| NTD-B7-WB | 25.00 ± 0.0 | | |
| NTD-O1-DCM | **87.50 ± 0.0** | 19.40 ± 2.8 | |
| NTD-O1-WB | 22.91 ± 2.9 | | |
| NTD-O2-DCM | **79.17 ± 5.9** | 39.10 ± 8.3 | |
| NTD-O2-WB | 20.83 ± 5.9 | | |
| NTD-O3-DCM | **100.00 ± 0.0** | 39.10 ±8.3 | |
| NTD-O3-WB | 8.34 ± 11.8 | | |
| NTD-O5-DCM | **93.75 ± 8.8** | 35.10 ± 2.8 | |
| NTD-O5-WB | 20.83 ± 5.9 | | |
| NTD-E1-DCM | **93.75 ± 8.8** | 33.20 ± | |
| NTD-E1-WB | 22.92 ± 2.9 | | |

* Calculated on adult activity. Experiments were done in triplicates and repeated once.

the traditional medicine practitioners, they utilize it as a preservative in the aqueous preparations.

In the primary screens, six extracts–NTD-B2-DCM, NTD-B3-DCM, NTD-B4-DCM, NTD-B5- BuOH, NTD-O5-DCM and NTD-E1-DCM—displayed 100% inhibition on *O. ochengi* microfilariae while four others–NTD-B3-MeOH, NTD-O1-DCM, NTD-O2-DCM and NTD-O3-DCM—were moderately active (50–75%) (Table 4). For the adult *O. ochengi* worms, 17 crude extracts–NTD-B1-DCM, NTD-B2-DCM, NTD-B3-DCM, NTD-B3-MeOH, NTD-B4-DCM, NTD-B4-MeOH, NTD-B5- DCM, NTD-B5-BuOH, NTD-B6-DCM, NTD-O6-BuOH, NTD-B7-DCM, NTD-B7-BuOH, NTD-O1-DCM, NTD-O2-DCM, NTD-O2-WB, NTD-O3-DCM and NTD-O4-DCM—exhibited a 100% inhibition of the male. Out of these 17 extracts, NTD-O1-DCM and NTD-B6-DCM were the most effective against the female adult worms, demonstrating 98.8% and 91.7% kill, respectively. Eight extracts, NTD-B2-DCM (66.3%), NTD-B3-DCM (57.3%), NTD-B7-DCM (63.8%), NTD-O2-DCM (61.0%), NTD-O2-WB (56.6%), NTD-O3-DCM (53.8%), NTD-O4-DCM (64.8%) and NTD-E1-DCM (53.8%) displayed moderate activity. In the secondary screening, only extracts which displayed 100% inhibition of male and >60% killing of the female worms were tested. Among the 4 extracts that qualified for this stage of testing, NTD-B2-DCM displayed the highest activity with an IC$_{50}$ of 76.7 µg/mL and 76.2 µg/mL, compared to 0.27 and 0.20 µg/mL for auranofin, against the male and female adult worms, respectively (Table 4).

**Table 4. Antionchocercal activity and cytotoxicity of crude extracts against *O. ochengi*.**

| Crude Extracts (Conc. 200 µg/mL) | Cytotoxic? | Microfilaria | Adult worms | | | *IC$_{50}$ µg/mL | |
|---|---|---|---|---|---|---|---|
| | Yes/No/ Moderate | % inhibition (5 days) | % inhibition of males (5 days) | % killing of females (7 days) | | Adult male | Adult female |
| NTD-B1-DCM | No | 25±0 | 100±0 | 22.6±2.5 | | | |
| NTD-B1-BuOH | No | 0 | 0 | 10.1±2.1 | | 76.7 | 76.2 |
| **NTD-B2-DCM** | **Yes** | **100±0** | **100±0** | **66.3±9.0** | | **100.0** | **109.0** |
| NTD-B2-BuOH | No | 0 | ND | 35.1±18 | | **154.0** | >200 |
| NTD-B3-DCM | Yes | 100±0 | 100±0 | 57.6±4.4 | | >200 | >200 |
| NTD-B3-MeOH | Yes | 75±0 | 100±0 | 41.3±18 | | | |
| NTD-B4-DCM | Yes | 100±0 | 100±0 | 47.6±0 | | | |
| NTD-B4-MeOH | No | 25±0 | 100±0 | 35.1±0 | | | |
| NTD-B5-DCM | No | 0 | 100±0 | 47.6±2.5 | | | |
| NTD-B5-BuOH | Moderate | 100±0 | 100±0 | 35.1±22 | | | |
| **NTD-B6-DCM** | **Yes** | **25±0** | **100±0** | **91.7±0** | | | |
| NTD-B6-BuOH | No | 0 | 100±0 | 22.6±0 | | | |
| **NTD-B7-DCM** | **No** | **0** | **100±0** | **63.8±4.4** | | | |
| NTD-B7 DCM/S1 | ND | 0 | 0 | 0 | | | |
| NTD-B7-BuOH | No | 0 | 100±0 | 41.3±11 | | | |
| **NTD-O1-DCM** | **Yes** | **50±0** | **100±0** | **98.8±1.8** | | **105.5** | **141.4** |
| NTD-O1-WB | No | 0 | 68.8±0 | 25.3±3.3 | | | |
| NTD-O2-DCM | No | 50±0 | 100±0 | 61.0±1.8 | | | |
| NTD-O2-WB | No | 0 | 100±0 | 56.6±4.4 | | | |
| NTD-O3-DCM | No | 75±0 | 100±0 | 53.8±4.4 | | | |
| NTD-O3-WB | No | 0 | 0 | 5.2±0 | | | |
| NTD-O4-DCM | No | 25±0 | 100±0 | 28.8±0 | | | |
| NTD-O4-MeOH | No | 25±0 | 6.3±1.4 | 6.3±0.4 | | | |
| NTD-O5-DCM | Yes | 100±0 | ND | 64.8±3.8 | | | |
| NTD-O5-WB | No | 0 | ND | 7.8±0 | | | |
| NTD-E1-DCM | Yes | 100±0 | ND | 53.8±18 | | | |
| NTD-E1-WB | No | 0 | 0 | 18.8±2.5 | | | |
| Auranofin (10 µM) | | 100 | 100 | 100 | | 0.27 | 0.20 |

* Calculated on adult activity. Experiments were done in quadruplicates for adult worms, duplicates for microfilariae, and repeated once.

The preliminary cytotoxicity profile of the crude extracts revealed that all the active fractions were also cytotoxic against the LLC-MK2 cells except NTD-B5-BuOH which was moderately cytotoxic. The moderately active extracts NTD-O2-DCM and NTD-O3-DCM were, however, not cytotoxic.

## Antitrypanosomal activity of the extracts

Eight out of the 26 crude extracts tested (NTD_B2-DCM, NTD_B4-DCM, NTD_O1-DCM, NTD_O2-WB, NTD_O3-DCM, NTD_O4-DCM, NTD_E1-DCM and NTD_E1-WB) exhibited good antitrypanosomal activity with IC$_{50}$ values ranging between 5 and 10 µg/mL (Table 5). The remaining 18 extracts possessed moderate antitrypanosomal activity with IC$_{50}$ values ranging from 10 µg/mL to 19 µg/mL. The IC$_{50}$ value of the standard drug, DA, was 0.13 µg/mL.

## Broad-spectrum activities of the extracts

Five extracts displayed activity in more than one biological assay (Table 6). Particularly, extracts prepared from the antischistosomal TMs had greater and broader spectrum of activities. NTD-B2-DCM was the most active antionchocercal extract with IC$_{50}$ values of 76.2 µg/mL and 76.7 µg/mL against the male and female adult worms, respectively, and also the second most active antitrypanosomal extract (IC$_{50}$ = 7.12 µg/mL). However, it did not show any activity against its target NTD, schistosomiasis. NTD-B4-DCM was the most active extract against

**Table 5. Antitrypanosomal activity of crude extracts against *T. brucei brucei*.**

| Crude Extracts | IC$_{50}$ (µg/mL) Mean ± SD |
|---|---|
| NTD-B1-DCM | 14.05 ± 1.54 |
| NTD-B1-BuOH | 11.40 ± 1.51 |
| **NTD-B2-DCM** | **7.12 ± 1.57** |
| NTD-B2-BuOH | 12.44 ± 1.87 |
| NTD-B3-DCM | 11.95 ± 1.12 |
| NTD-B3-MeOH | 18.79 ± 1.33 |
| **NTD-B4-DCM** | **5.63 ± 0.89** |
| NTD-B4-MeOH | 16.80 ± 6.68 |
| NTD-B5-DCM | 10.89 ± 2.20 |
| NTD-B5-BuOH | 14.19 ± 7.26 |
| NTD-B6-DCM | 13.67 ± 1.08 |
| NTD-B6-BuOH | 14.34 ± 0.97 |
| NTD-B7-DCM | 10.88 ± 1.42 |
| NTD-B7-BuOH | 16.10 ± 3.88 |
| NTD-O1-WB | 10.84 ± 1.54 |
| **NTD-O1-DCM** | **8.92 ± 3.79** |
| NTD-O2-DCM | 10.68 ± 2.54 |
| **NTD-O2-WB** | **9.44 ±1.88** |
| **NTD-O3-DCM** | **7.27 ± 1.59** |
| NTD-O3-WB | 16.41 ± 2.49 |
| **NTD-O4 DCM** | **8.69 ± 1.55** |
| NTD-O4 MeOH | 10.78 ± 1.49 |
| **NTD-O5-WB** | **10.40 ± 0.97** |
| **NTD-O5-DCM** | **10.10 ± 1.01** |
| **NTD-E1-DCM** | **9.55 ± 3.49** |
| **NTD-E1-WB** | **7.17 ± 0.88** |
| Diminazene aceturate | 0.13 ± 0.02 |

Means are averages of three biological replicates. Each biological replicate had three technical replicates.

two of the diseases, its target NTD schistosomiasis (IC$_{50}$ = 30.5 µg/mL) and trypanosomiasis (IC$_{50}$ = 5.63 µg/mL). The extract with the second highest antischistosomal activity was NTD-B7-DCM (IC$_{50}$ = 30.8 µg/mL) which further recorded an antitrypanosomal activity of IC$_{50}$ = 10.88 µg/mL. Both NTD-B4-DCM and NTD-B7-DCM were, however, not effective against *O. ochengi* female worms. Generally, the antionchocercal remedies were less effective against the adult-stage lifecycle forms of *S. mansoni* and *O. ochengi*, and therefore did not qualify for secondary screening. NTD-O1-DCM exhibited the highest percentage inhibition (100%) of *O. ochengi* male and percentage killing (98.8%) of the female worms, but recorded IC$_{50}$ values >100 µg/mL. On the other hand, the antionchocercal remedies showed good antitrypanosomal activities, with the lowest IC$_{50}$ value (8.92 µg/mL) recorded by NTD-O1-DCM. NTD-E1-DCM, prepared from an antiLF traditional medicine, showed good antitrypanosomal activity (IC$_{50}$ = 9.55 µg/mL), was highly effective against *S. mansoni* NTS (94%) and *O. ochengi* microfilariae (100%) but was inactive against the corresponding adult worms.

## Bioactivity-guided fractionation of NTD-B4-DCM

Since the antitrypanosomal assay of the extracts against the bloodstream forms of *T. b. brucei* gave the most promising results (IC$_{50}$ = 5.63 µg/mL to 18.79 µg/mL) when compared with the

**Table 6. Summary of extracts with broad spectrum activities.**

| Crude Extracts | IC$_{50}$ (µg/mL) | | | |
| --- | --- | --- | --- | --- |
| | Antischistosomal activity | Antionchocercal activity | | Antitrypanosomal activity |
| | | male | female | |
| NTD-B2-DCM | NA | 76.7 | 76.2 | 7.12 |
| NTD-B4-DCM | 30.5 | NA | NA | 5.63 |
| NTD-B6-DCM | NA | 100.0 | 109.0 | 13.67 |
| NTD-B7-DCM | 30.8 | 154.0 | >200 | 10.88 |
| NTD-O1-DCM | NA | 105.5 | 141.4 | 8.92 |

NA: not active. Values shown are mean for 3 determinations. For all the assays, extracts with IC$_{50}$ values below 10 µg/mL are considered to possess good biological activity.

other screens, this assay was selected for bioactivity-guided fractionation of the most active extract. NTD-B4-DCM, the most active crude extract, displayed the highest antitrypanosomal activity (IC$_{50}$ = 5.63 µg/mL) of all the extracts tested. Flash chromatography of NTD-B4-DCM yielded 6 fractions—F1 to F6, all of which were screened for antitrypanosomal activity (Table 7).

F5 was the most active fraction (IC$_{50}$ = 7.37 µg/mL), followed by F1 (IC$_{50}$ = 8.50 µg/mL). The least active fraction was F2 (IC$_{50}$ = 64.47 µg/mL) while F6 was about twice as active as F2 (IC$_{50}$ = 30.62 µg/mL). Although all the fractions were overall less active than the crude extract from which they were obtained, NTD-B4-DCM (IC$_{50}$ = 5.63 µg/mL), and the standard (IC$_{50}$ = 0.13 µg/mL); nonetheless, further chromatographic separation was pursued in order to isolate the active ingredient(s) responsible for the observed antitrypanosomal activity.

Due to paucity of F5, the next most active fraction F1, was selected for chemical profiling which led to the isolation of 2 solid compounds (F1/K and F1/J) and 3 oils (F1/JML, F1/HML and F1/KML). Following evaluation of the isolates and other eluents from the chromatographic separation for antitrypanosomal activity, all the compounds were not significantly active against the trypanosomes (F1/J, F1/B, F1/F, F1/H, F1/S and F1/Z: IC$_{50}$ >100 µg/mL; F1/K: 80.95 µg/mL) (Table 8). On the other hand, all the oils, apart from F1/JML (IC$_{50}$ = 32.20 µg/mL), exhibited the promising antitrypanosomal activity (Table 8). F1/HML, in particular, had an antitrypanosomal activity of <0.0977 µg/mL, which is about 1.3-fold greater than the standard control drug used, DA (IC$_{50}$ = 0.13 µg/mL). Preliminary cytotoxicity tests revealed that F1/HML is non-toxic to murine macrophages cells (RAW 264.7) and this indicates that the oil would selectively kill trypanosomes, sparing host cells.

## Structure elucidation of F1/K

Compound F1/K was isolated as a pale-yellow powder and showed a molecular formula of $C_{24}H_{36}O_4$ based on its molecular ion peak at m/z 388.3938 [M$^+$] in the HR-ESIMS (S1 Fig).

**Table 7. Antitrypanosomal activity of fractions of NTD-B4-DCM.**

| NTD-B4-DCM Fractions | IC$_{50}$ (µg/mL) Mean ± SD |
| --- | --- |
| **F1 (100% PE + 10% EtOAc)** | **8.50 ± 0.48** |
| F2 (20% EtOAc) | 64.47 ± 7.20 |
| F3 (50% EtOAc) | 19.96 ± 0.12 |
| F4 (80% EtOAc) | 11.61± 2.86 |
| **F5 (100% EtOAc)** | **7.37 ± 0.83** |
| F6 (EtOAc/MeOH; 1:1) | 30.62 ± 0.95 |
| Diminazene aceturate | 0.13 ± 0.02 |

**Table 8. Antitrypanosomal activity of compounds and other constituents of fraction F1.**

| Sample Code | Nature | IC$_{50}$ (µg/mL) Mean ± SD |
| --- | --- | --- |
| F1/B | - | >100 |
| F1/F | - | 49.51 ± 4.18 |
| F1/H | - | >100 |
| **F1/HML** | **oil** | **<0.0977** |
| F1/J | solid | >100 |
| F1/JML | oil | 32.20 ± 1.72 |
| F1/K | solid | 80.95 ± 4.15 |
| **F1/KML** | **oil** | **8.06 ± 0.38** |
| F1/S | - | >100 |
| F1/Z | - | >100 |
| Diminazene aceturate | | 0.13 ± 0.02 |

The presence of hydroxyl (3394 cm$^{-1}$), α, β-unsaturated carbonyl (1646 cm$^{-1}$) and aromatic (1629.69 cm$^{-1}$) functionalities was evident from the IR absorption spectrum. The $^{13}$C NMR spectrum (S2 Fig) exhibited 16 signals which were established by the Distortionless Enhancement by Polarization Transfer (DEPT) experiment as seven quaternary, five methylene, two methine and two methyl carbons. Resonances included those of a carbonyl C-4 ($\delta_C$ 182.6), two olefinic carbons at $\delta_C$ 170.4 (C-2) and $\delta_C$ 108.0 (C-3), six aromatic carbons and an aromatic methyl (C-11) at $\delta_C$ 7.1. A long signal at $\delta_C$ 29.8 was characteristic of an aliphatic carbon chain bearing a terminal methyl at $\delta_C$ 14.2 (C-25). The $^1$H NMR spectrum (S3 Fig) revealed a sharp singlet at $\delta_H$ 12.92 assigned to the hydroxyl protons, an aromatic proton H-8 ($\delta_H$ 6.33, s), an olefinic proton H-3 ($\delta$H 6.01, s) and an intense absorption peak at $\delta_H$ 1.26 which corroborated the aliphatic nature of the compound. Correlation Spectroscopy (COSY) cross-peaks were observed in the alkyl chain between H-12/H-13, H-13/H-14 and H-24/H-25. Heteronuclear Multiple Bond Correlation Spectroscopy (HMBC) correlations from OH to C-5 ($\delta_C$ 159.7), C-7 ($\delta_C$ 160.0), C-6 ($\delta_C$ 106.9) and C-10 ($\delta_C$ 103.4), H-8 to C-6 and C-10, and H-3 to C-10 and the oxygenated quaternary carbon C-2 ($\delta_C$ 170.4) were supportive of a dihydroxy-substituted chromenone derivative. The upfield chemical shift of the carbonyl C-4 ($\delta_C$ 182.6) confirmed the existence of an α, β-unsaturated ketone; hence a 4-chromenone nucleus. The position of the aromatic methyl was established by the correlation of H-11 ($\delta_H$ 2.31, s) to C-6 while further correlations from H-12 ($\delta_H$ 2.55, J = 2.2 Hz) to C-2 ($\delta_C$ 170.4) and H-3 ($\delta_H$ 6.01, s) to C-12 ($\delta_C$ 34.3) revealed the connection of the aliphatic sidechain to the 4-chromenone nucleus. Thus, F1/K was identified as 5,7-dihydroxy-6-methyl-2-tetradecyl-4*H*-chromen-4-one in support of the molecular formula, C$_{24}$H$_{36}$O$_4$ (Fig 1).

Similar 4-chromenone compounds constituting higher homologues of F1/K have been previously isolated by Cooke and Down [43], from the plant species *Stypandra gradis* and *Daniella*

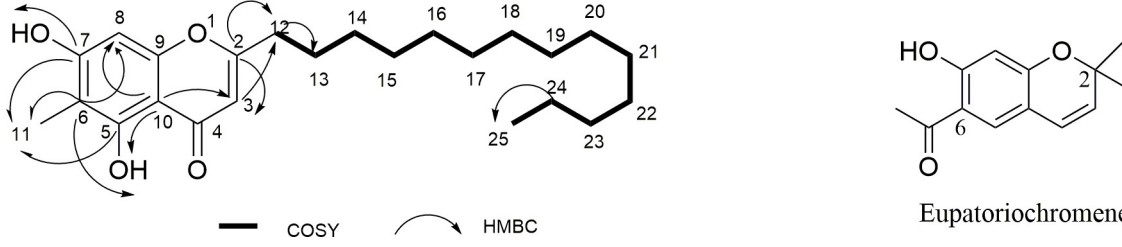

**Fig 1. Structures of compound F1/K and Eupatoriochromene.**

*revolta* with an attempted synthesis and proposed biosynthetic pathway for these compounds. The major isolated compound in their study, with molecular formula $C_{39}H_{66}O_4$ (*m/z* 598), was accompanied by the homologues $C_{37}H_{62}O_4$ (*m/z* 570) and $C_{41}H_{70}O_4$ (*m/z* 626). The biological activities of the compounds were however, not reported. Generally, 4-chromenone derivatives are an important class of naturally occurring compounds and considered as privileged scaffolds in chemotherapeutics due to their wide spectrum of biological activities which include kinase inhibitory [44], anti-fungal [45], inflammatory [46], antioxidant [47] and anti-cancer and antibacterial [48].

### GCMS analysis of F1/HML

F1/HML was obtained as a viscous bright orange oil whose GC-MS analysis revealed its major constituent (42.6%) as a chromene derivative, 1-(7-hydroxyl-2,2-dimethyl-2*H*-chromen-6-yl)-ethanone, also known as eupatoriochromene (Fig 1). The remaining constituents of F1/HML, including 2 sesquiterpenoids, are presented in Table 9 and S4 and S5 Figs.

## Discussion

In this study, fifteen traditional medicines used as remedies for schistosomiasis, onchocerciasis and LF in Ghanaian communities were evaluated for activity, firstly against their target diseases and then, in all other available screening platforms for other NTDs in order to detect broad-spectrum activity. The assays that were obtained for testing were for schistosomiasis (*S. mansoni*), onchocerciasis (*O. chengi*) and African trypanosomiasis (*T. b. brucei*). No antiLF assay was accessible; hence the LF remedies were not tested against their target disease. African Trypanosomiasis affects humans (HAT) and animals (AAT), causing severe disease. HAT is an NTD, though there has been a decline in the number of reported cases in recent times. AAT, however, is on the rise and poses a threat that might trigger a re-emergence of HAT if not properly controlled. All the extracts prepared from the traditional medicines were tested for activity against schistosomiasis, onchocerciasis and African trypanosomiasis.

The two most active extracts in the antischistosomal screening, NTD-B4-DCM ($IC_{50}$ = 30.5 µg/mL) and NTD-B7-DCM ($IC_{50}$ = 30.8 µg/mL), were prepared from TMs used locally for treating schistosomiasis. NTD-B4 is administered orally as an infusion of one teaspoon of the dried herbs of *Aloe vera* and *Taraxacum officinale* in 100 mL of warm water three times daily. Feedback from the TM practitioner did not indicate how many people have been treated after using the remedy. According to the practitioners, successful treatment is assessed by the reduction or complete relief of symptoms. For instance, in the case of schistosomiasis which may be characterized by bloody urine or diarrhoea, the loss of these symptoms after taking the remedy, constitutes a positive outcome. For NTD-B4, the DCM extract was more active than

**Table 9. Compounds identified in F1/HML by GC-MS analysis.**

| Retention time (min) | Compound | Molecular formula | Molecular mass (Da) | Composition (%) |
|---|---|---|---|---|
| 47.267 | geranyl acetone | $C_{13}H_{22}O$ | 194.31 | 4.14 |
| 50.734 | unidentified | - | - | 16.19 |
| 55.525 | caryophyllene oxide | $C_{15}H_{24}O$ | 220.35 | 13.80 |
| 57.152 | humulene epoxide II | $C_{15}H_{24}O$ | 220.35 | 6.95 |
| 66.305 | **eupatoriochromene** | $C_{13}H_{14}O_3$ | 218.25 | **42.60** |
| 69.929 | phytone | $C_{18}H_{36}O$ | 268.50 | 2.75 |
| 73.062 | 5Z,9E-farnesyl acetone | $C_{18}H_{30}O$ | 262.43 | 11.89 |
| 73.808 | methyl palmitate | $C_{17}H_{34}O_2$ | 270.45 | 1.68 |

the MeOH extract (Table 3), suggesting that the active principles were extracted into the moderately polar DCM before the subsequent introduction of the polar solvent, MeOH. NTD-B7 is an aqueous-based remedy prepared from four plant species (*Vernonia amygdalina*, *Khaya senegalensis*, *Mangifera indica*, and *Azadirachta indica*) and the dosage regimen is five tablespoons three times daily. Similarly, the DCM extract was the active extract. According to the TM practitioner, >20 patients have benefitted from the use of NTD-B7 against schistosomiasis. The protocol employed in the antischistosomal assay [38] requires that compounds that show activity with $IC_{50}$ values less than 10 μM in the *in vitro* adult-stage worm screens can progress into *in vivo* testing. Although the two extracts did not exhibit this level of activity, they have been prioritized for bioassay-guided fractionation to ascertain if purification would lead to isolation of molecules that possess enhanced activity than the crude extracts.

In the antionchocercal primary screens, activity against the female worm is considered the most important criterion for selection into follow-up testing because this stage of the lifecycle has proven the hardest to kill and it is also the reproductive stage that gives rise to the microfilariae [39]. As shown in Table 4, majority of the extracts exhibited high to moderate inhibition of *O. ochengi* microfilariae and the adult male worm primary screens. All the eleven DCM extracts screened against the male worm displayed 100% inhibition while the remaining two (NTD-O5-DCM and NTD-E1-DCM) which were not tested against the male worms, also completely inhibited the microfilariae. This observation corroborates other studies where filaricidal activities of plant extracts have been found to reside in the nonpolar fractions [49,50]. However, the outcome of the adult female worm screening was less favourable–only 6 extracts, all from DCM, exhibited >60% inhibition. The four extracts that eventually qualified for secondary screens exhibited low antionchocercal activity. The most active extract, NTD-B2-DCM ($IC_{50}$ = 76.2 μg/mL and 76.7 μg/mL for female and male, respectively), was obtained from the DCM extract of an antischistosomal TM prepared from a combination of *Syzygium aromaticum*, *Xylopia aethiopica*, *Tapinanthus bangwensis* and *Phyllanthus niruri*). Incidentally, this extract was not active against its target disease, schistosomiasis, suggesting the need to screen TMs for activity against multiple pathogens before refuting their local use. Three other DCM extracts, prepared from two antischistosomal (NTD-B6 and NTD-B7) and one antionchocercal (NTD-O1) remedies, recorded $IC_{50}$ values of ≥ 100 μg/mL. NTD-O1 is an aqueous preparation from *Mangifera indica*, *Momordica charantia*, *Zingiber officinale* and *Xylopia aethiopica*, of which two tablespoons are taken three times a day for onchocerciasis. It is purported by the practitioner that the use of the remedy has resulted in the treatment of 15–20 people (Table 1). Interestingly, in the primary screen, NTD-O1-DCM was the most effective against the female adult worm, demonstrating 98.8% kill (Table 4). This result could be responsible for the therapeutic claim of the remedy. Coupled with the low antionchocercal activities, the extracts were also cytotoxic against Monkey Kidney Epithelial (LLC-MK2) cells from the preliminary assay (Table 4), indicating the need for practitioners to adhere to proper standardization protocols including proven safety, regulated dose regimens and good manufacturing practices.

It is noteworthy that the overall best performance of the extracts was observed in the antitrypanosomal assay, although, none of the twenty-six extracts which displayed good to moderate antitrypanosomal activity ($IC_{50}$ = 5 μg/mL to 19 μg/mL; Table 5), was purposed for use against African trypanosomiasis. Extracts NTD-B4-DCM and NTD-B2-DCM, prepared from antischistosomal remedies, recorded the lowest $IC_{50}$ values of 5.63 and 7.12 μg/mL, respectively. They were followed closely by the antiLF remedy, NTD-E1-WB ($IC_{50}$ = 7.17 μg/mL) and the antionchocercal remedy NTD-O3-DCM ($IC_{50}$ = 7.27 μg/mL). Coincidentally, NTD-B4-DCM and NTD-B2-DCM were also the most active samples in the antischistosomal and antionchocercal assays, respectively, highlighting the broad-spectrum activity of these

extracts. While no single extract was active in all the three screening platforms, three additional extracts showed activity in two biological assays. NTD-B7-DCM was nearly as active as NTD-B4-DCM in the antischistosomal assay ($IC_{50}$ = 30.8 μg/mL) but its antitrypanosomal activity was about twice less ($IC_{50}$ = 10.88 μg/mL). NTD-B6-DCM and NTD-O1-DCM recorded $IC_{50}$ values of 13.67 μg/mL and 8.92 μg/mL, respectively in the antitrypanosomal assay but exhibited low anti-onchocercal activity ($IC_{50} \geq$ 100 μg/mL). From the observed broad-spectrum activities of some of the extracts (Table 6), it seems reasonable to deduce that this could be due to the presence of different classes of compounds with different mechanisms of action. One important finding from this study is that there are instances where TMs do not have any significant activity against their target disease but might be beneficial in the treatment of other NTDs. This observation highlights the need for comprehensive studies that validate or refute the local use of TMs.

The promising outcome of the evaluation of the activity of the extracts against *T. b. brucei* prompted us to pursue the bioactive antitrypanosomal constituents of the most active extract NTD-B4-DCM, via bioassay-guided fractionation. While fractionation of the crude extract resulted in less active fractions (Table 7), chromatographic separation of fraction F1 yielded a potent oil whose activity was 1.3-fold ($IC_{50}$ <0.0977 μg/mL) greater than that of the standard antitrypanosomal drug, diminazene aceturate ($IC_{50}$ = 0.13 μg/mL). Preliminary cytotoxicity tests revealed that the oil is non-toxic to murine macrophages cells (RAW 264.7) and therefore shows a lot of promise as a new antitrypanosomal agent. Eupatoriochromene (Fig 1, Table 9), identified as the major constituent (42.6%) of the oil, has previously been found as a minor component (<1%) of essential oils obtained by hydrodistillation of *Beilschmiedia* species (Lauraceae), used in traditional medicine for tumour, diarrhoea, rubella and wound healing [51]. Eupatoriochromene has also been isolated together with structurally related compounds from *Tithonia diversifolia* (Compositae), a perennial herb widely valued in several cultures for its medicinal properties [52–54]. Olukunle et al. [55] found that a 3-day administration of an aqueous leaf extract of *T. diversifolia* (400 mg/kg per day) to rats infested with *T. b. brucei* resulted in about 50% reduction of parasitaemia, suggesting that the plant is endowed with antitrypanosomal properties. The high antiprotozoal activity of natural products possessing chromane and chromene scaffolds inspired the synthetic work of Harel et al. [56] which unraveled a new class of potent antitrypanosomal ($IC_{50}$ = 1.03 and 1.84 μM), antileishmanial ($IC_{50}$ = value of 0.57 μM) and antiplasmodial ($IC_{50}$ = 0.02 μM) compounds. Eupatoriochromene served as a key intermediate for the synthesis of the various chromane and chromene analogues. Based on these findings, it can be suggested that the presence of eupatoriochromene in the oil isolated in the current study, might be a contributory factor for the observed potent antitrypanosomal activity. The 4-chromenone derivative, on the hand, exhibited low antitrypanosomal activity when tested ($IC_{50}$ = 80.95 μg/mL). Although there was no identified published work for comparative analysis, it is possible that structural effects, particularly, its long alkyl sidechain compared with eupatoriochromene (Fig 1), accounted for the observed diminished activity. The outcome of the bioassay-guided fractionation of NTD-B4-DCM provides a good basis for subjecting the remaining extracts with $IC_{50}$ < 10 μg/mL to the same process. This will help establish the effect of purification on activity and toxicity of the crude TMs with respect to their fractions and constituent compounds.

## Conclusion

The outcomes of the study suggest that traditional medicines used in treating neglected tropical diseases in Ghana hold promise as phytomedicines not only for their target diseases but for other NTDs as well. Undoubtedly, efficacy, safety and quality tests are warranted to validate the claims of traditional medicine practitioners. By embracing indigenous knowledge systems

which have evolved over centuries, we can potentially unlock a wealth of untapped research and shape it by conducting sound scientific investigations to produce safe, efficacious and good quality remedies. In terms of research and development, this model offers a real opportunity for 'African solutions to African problems'. Concomitantly, through bioassay-guided fractionation protocols and chemical profiling, traditional medicines should be explored as sources of lead compounds for drug discovery and development for NTDs.

## Supporting information

**S1 Fig. HRESIMS of F1/K (5,7-dihydroxy-6-methyl-2-tetradecyl-4H-chromen-4-one).** (TIF)

**S2 Fig. $^{13}$C NMR spectrum of F1/K.** (TIF)

**S3 Fig. $^{1}$H NMR spectrum of F1/K.** (TIF)

**S4 Fig. GC fingerprint of F1/HML.** (TIF)

**S5 Fig. Mass spectrum of eupatoriochromene.** (TIF)

## Acknowledgments

We wish to thank Dr. Anastasia Yirenkyi, Director of the Traditional and Alternative Medicine Directorate, Ministry of Health, Ghana, for facilitating the initial meetings with the Executives of the Ghana Federation of Traditional Medicines Practitioners Association (GHAFTRAM). We also wish to thank the Executives and members of GHAFTRAM for graciously providing the traditional medicines for screening, particularly those who had to submit additional samples for follow-up screening.

## Author Contributions

**Conceptualization:** Kwaku Kyeremeh, Richard K. Amewu, Dorcas Osei-Safo.

**Formal analysis:** Emmanuella Bema Twumasi, Pearl Ihuoma Akazue, Theresa Manful Gwira, Jennifer Keiser, Fidelis Cho-Ngwa, Dorcas Osei-Safo.

**Funding acquisition:** Kwaku Kyeremeh, Adrian Flint, Richard K. Amewu, Linda Eva Amoah, Regina Appiah-Opong, Dorcas Osei-Safo.

**Investigation:** Emmanuella Bema Twumasi, Pearl Ihuoma Akazue, Kwaku Kyeremeh, Theresa Manful Gwira, Jennifer Keiser, Fidelis Cho-Ngwa, Barbara Anibea, Emmanuel Yeboah Bonsu, Dorcas Osei-Safo.

**Resources:** Kwaku Kyeremeh, Theresa Manful Gwira, Jennifer Keiser, Fidelis Cho-Ngwa, Dorcas Osei-Safo.

**Supervision:** Kwaku Kyeremeh, Theresa Manful Gwira, Dorcas Osei-Safo.

**Writing – original draft:** Emmanuella Bema Twumasi, Dorcas Osei-Safo.

**Writing – review & editing:** Pearl Ihuoma Akazue, Kwaku Kyeremeh, Theresa Manful Gwira, Jennifer Keiser, Fidelis Cho-Ngwa, Adrian Flint, Linda Eva Amoah.

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
