## [Decision Letter · Decision Letter 0]

14 Jun 2020

Dear Prof Osei-Safo,

Dear Dr. Dorcas Osei-Safo and collegues.

Thank you very much for submitting your manuscript "Antischistosomal, antionchocercal
and antitrypanosomal potentials of some Ghanaian traditional medicines and their
constituents" for consideration at PLOS Neglected Tropical Diseases. As with all
papers reviewed by the journal, your manuscript was reviewed by members of the
editorial board and by several independent reviewers. In light of the reviews (below
this email), we would like to invite the resubmission of a significantly-revised
version that takes into account the reviewers' comments. 

Whereas the topic is highly important and the helminth community needs to fill up the
empty drug development pipe lines by all means, describing tests on herbal remedies
without naming the plant sources is an issue. Acknowledging that traditional
practitioners are reluctant to release information, it would be an asset for this
paper to add this information. 

We cannot make any decision about publication until we have seen the revised
manuscript and your response to the reviewers' comments. Your revised manuscript is
also likely to be sent to reviewers for further evaluation.

Sincerely,

Sabine Specht

Associate Editor

Francesca Tamarozzi

Deputy Editor

Dear Dr. Dorcas Osei-Safo and collegues.

Thanks for submitting your manuscript to PLoSNTD. Whereas the topic is highly
important and the helminth community needs to fill up the empty drug development
pipe lines by all means, describing tests on herbal remedies without naming the
plant sources is an issue. Acknowledging that traditional practitioners are
reluctant to release information, it would be an asset for this paper to add this
information. 

With best regards, Sabine Specht

Reviewer's Responses to Questions

**Key Review Criteria Required for Acceptance?**

**Methods**

-Are the objectives of the study clearly articulated with a clear testable hypothesis
stated?

-Is the study design appropriate to address the stated objectives?

-Is the population clearly described and appropriate for the hypothesis being
tested?

-Is the sample size sufficient to ensure adequate power to address the hypothesis
being tested?

-Were correct statistical analysis used to support conclusions?

-Are there concerns about ethical or regulatory requirements being met?

Reviewer #1: The methods are standard, but are not adequately described.

The main objection is that the material studied (herbal remedies) is not
described.

Reviewer #2: 1) Table 1, number of people successfully treated is not an appropriate
heading since you said earlier that ‘According to the practitioners, their efficacy
claims are based on the number of people who experience reduction or relief from
symptoms as a result of using their products’. So it should be coined as the number
of people relieved of symptoms and not successfully treated. Given that no
laboratory tests were done to determine parasite loads, it is subjective to conclude
that the patients were treated successfully

2) Statistical analysis for comparisons of different extracts and concentrations for
onchocerca and schistosoma will enhance the credibility of findings given that you
ran the assays in quadruplicates and triplicates repeatedly.This will reveal if the
differences observed are statistically significant in different concentrations or at
the IC50. A good example is Table 5.

Minor Points

3) We recommend that the quality of traditional medicines be improved (Author
summary)

4) Primary screening of adult worms and microfilariae, Secondary screening on adult
worms

It should be “screening against adult worms” instead of “screen of adult worms”

5) Correct from GCMSNMR to GC/MS/NMR in the manuscript as they are different
techniques. 

6) Discussion section, ‘however, is on the rise and possess’ it should be poses

7) Introduction section ‘high cost of bringing new drugs to the market’

Reviewer #3: (No Response)

**Results**

-Does the analysis presented match the analysis plan?

-Are the results clearly and completely presented?

-Are the figures (Tables, Images) of sufficient quality for clarity?

Reviewer #1: Some of the figures are illegible, or barely legible.

Reviewer #2: 1) Cytotoxicity assessment of extracts. The 50% cytotoxic concentrations
(CC50) of the cells were determined by microscopic examinations. 

Looking at Table 4, the cytotoxicity was done at 200 µg/mL only. So the CC50 was not
determined as stated in the methodology. The selectivity index (SI) column should be
included if the CC50 was determined. Also, consider including this aspect in your
abstract

Reviewer #3: The objectives of the study are clearly defined, but without a clear
testable hypothesis stated. 

An in vitro screening was used to evaluate the efficacy of 15 TM against NTDs. This
is an appropriate design for screening the extracts of the various TMs for activity
against NTD and evaluating their efficacy. The extracts were also assessed for
cytotoxicity against Monkey Kidney Epithelial (LLC-MK2) cells to ensure the safety
of the products. 

The protocols employed in the evaluation of the activity of the crude extracts of the
TMs were appropriate. 

Though the statistical analyses were no elaborate, the analyses carried out were
sound to provide a good interpretation of the data.

Also, ethical approval was obtained from the appropriate Institutional Review
Board

**Conclusions**

-Are the conclusions supported by the data presented?

-Are the limitations of analysis clearly described?

-Do the authors discuss how these data can be helpful to advance our understanding of
the topic under study?

-Is public health relevance addressed?

Reviewer #1: Yes. If anything, more than is necessary.

Reviewer #2: Looking at the results you obtained for the antionchocercal activity of
extracts, all the active extracts are dichloromethane extracts. Most filaricidal
activities of plants tested come from non-polar extracts. This could be backed up in
the discussion section with publications from other authors who had similar
findings.

Reviewer #3: The results and data support the conclusions, the discussion was well
written and authors clearly indicated how these data can be helpful to advance our
understanding of traditional medicine and its benefit the treatment of various
diseases including NTDs

The public health relevance of the study is clearly indicated and the authors have
emphasized the benefit of traditional medicine and how it could be explored for drug
discovery and development for NTDs

**Editorial and Data Presentation Modifications?**

Reviewer #1: (No Response)

Reviewer #2: Minor revision

Reviewer #3: Minor revision is required before manuscipt is accepted for
publication

**Summary and General Comments**

Reviewer #1: If the authors are able to publish the plant species, and present the
Methods and Results correctly, then the financial cost of this project would not be
wasted. There is no significant problem with the actual writing of English.

Reviewer #2: Onchocerciasis, schistosomiasis and lymphatic filariasis feature among
the most prevalent NTDS and are of top priority to elimination programs. The
Expanded Special Programme for Elimination of Neglected Tropical Diseases (ESPEN)
recently promotes an integrated approach to eliminating Neglected Tropical Diseases
and its strategy is optimisation and expansion of existing drugs to hypo-endemic
areas, plus potential integration of alternative strategies.

This manuscript by Twumasi et al fits into the goals of ESPEN and addresses key
issues that affect Africa and is of great significance. They evaluated the
‘antischistosomal, antionchocercal, and antitrypanosomal potentials of some Ghanaian
traditional medicines and their constituents’ of fifteen traditional medicines in
vitro. They carried out cytotoxicity assessment of the extracts and bioassay-guided
fractionation of the most active antitrypanosomal extract which yielded an oil
(eupatoriochromene). 

The manuscript is written in good english and the methodology and techniques are
appropriate. However there are some issues in the statistical analysis and
cytotoxicity data that needs to be adjusted.

Reviewer #3: The authors emphasized the need for the developments of new drugs for
Neglected Tropical Diseases (NTDs) to facilitate the elimination and eradication of
these diseases. Some current NTD drugs have shown reduced efficacy, serious adverse
effects associated treatment, while others are unable to prevent reinfection.
However, the development of new drugs for NTDs is very slow because of limited
investments and minimal interest by major drug companies. Fortunately, most African
countries including Ghana where NTDs are highly prevalent, abounds in indigenous
medicinal plant species whose products serve as primary sources of remedies for most
affected populations. 

This manuscript therefore addresses an interesting and important topic that provides
useful evidence for development of Traditional medicine and could potentially lead
to policy directions for the development of traditional medicines (TMs). However,
these resources are still deployed and practice in Indigenous settings. Thus, TMs
are underdeveloped and require significant research and development support to
harness the benefits that can be derived.

The authors investigated extracts of fifteen traditional medicines used for treatment
of NTDs including schistosomiasis, onchocerciasis and lymphatic filaliasis (LF) and
found some extracts to be efficacious against the causative agents of
schistosomiasis and onchocerciasis, however, no screening platform was found for LF.
The evaluation of the activity of the extracts showed that some TMs were very active
against the juvenile and adults of Schistoma mansoni as well as microfilaria and
adult male and female of Onchocerca ochengi. Surprisingly, some of the TMs which
were not indicated for African trypanosomiasis rather demonstrated the highest
activity against Trypanosoma brucei brucei. The authors have demonstrated the need
to evaluate for activity of various TMs against other diseases including NTDs to
determine the broad-spectrum activity of TM products.

The authors should have explored more options of collaboration to access more
screening platforms to test for activity against LF, especially since LF is one of
the NTDs in Ghana that have made significant progress towards elimination, but
currently facing challenges of advancing towards elimination. LF requires therefore
additional tools for eventual elimination and exploring effective TMs will be an
advantage. 

The findings of this study demonstrates the significant role of traditional medicines
and the prospects towards drug discovery and alternate medicine for NTDs and other
diseases.

PLOS authors have the option to publish the peer review history of their article
(what does this mean?). If published, this will
include your full peer review and any attached files.

If you choose “no”, your identity will remain anonymous but your review may still be
made public.

**Do you want your identity to be public for this peer review?** For
information about this choice, including consent withdrawal, please see our
Privacy Policy.

Reviewer #1: Yes: Jonathan A. Coles

Reviewer #2: No

Reviewer #3: No

Figure Files:

Data Requirements:

Reproducibility:

Editors.docx
to the Authors.docx
Review comments PLOS 1.doc
---

## [Decision Letter · Decision Letter 1]

15 Oct 2020

Dear Prof Osei-Safo,

Thank you very much for submitting your manuscript "Antischistosomal, antionchocercal
and antitrypanosomal potentials of some Ghanaian traditional medicines and their
constituents" for consideration at PLOS Neglected Tropical Diseases. As with all
papers reviewed by the journal, your manuscript was reviewed by members of the
editorial board and by several independent reviewers. The reviewers appreciated the
attention to an important topic. Based on the reviews, we are likely to accept this
manuscript for publication, providing that you modify the manuscript according to
the review recommendations. 

Sincerely,

Sabine Specht

Associate Editor

Francesca Tamarozzi

Deputy Editor

Reviewer's Responses to Questions

**Key Review Criteria Required for Acceptance?**

**Methods**

-Are the objectives of the study clearly articulated with a clear testable hypothesis
stated?

-Is the study design appropriate to address the stated objectives?

-Is the population clearly described and appropriate for the hypothesis being
tested?

-Is the sample size sufficient to ensure adequate power to address the hypothesis
being tested?

-Were correct statistical analysis used to support conclusions?

-Are there concerns about ethical or regulatory requirements being met?

Reviewer #2: Yes

Reviewer #3: The objectives of the study were clearly defined with clear hypothesis
stated

The methodologies used were appropriate to achieve the objectives of the study. The
in vitro assays used to assessed the efficacy of the 15 TMs against NTDs were
appropriately designed. Assessing the cytotoxicity of extracts using Monkey Kidney
Epithelial (LLC-MK2) cells was an appropriate method to evaluate the safety of the
extracts. 

Though the statistical analyses were not elaborate, the analyses carried out were
sound to provide a good interpretation of the data.

Also, ethical approval was obtained from the appropriate Institutional Review
Board

The methodologies used were appropriate to achieve the objectives of the study. The
in vitro assays used to assessed the efficacy of the 15 TMs against NTDs were
appropriately designed. Assessing the cytotoxicity of extracts using Monkey Kidney
Epithelial (LLC-MK2) cells was an appropriate method to evaluate the safety of the
extracts. 

Though the statistical analyses were not elaborate, the analyses carried out were
sound to provide a good interpretation of the data.

Also, ethical approval was obtained from the appropriate Institutional Review
Board

**Results**

-Does the analysis presented match the analysis plan?

-Are the results clearly and completely presented?

-Are the figures (Tables, Images) of sufficient quality for clarity?

Reviewer #2: Yes

Reviewer #3: The results presented are in line with the methods employed and the
analysis carried out by the authors. The evaluation of antischistosomal,
antionchocercal and antitrypanosomal activities of extracts showed that some TMs
were very active against the juvenile and adults of Schistoma mansoni as well as
microfilaria and adult male and female of Onchocerca ochengi, while some of these
extracts demonstrated the highest activity against Trypanosoma brucei brucei, the
causative agents of African Animal Trypanosomiasis, The authors have demonstrated
the need to evaluate for activity of various TMs against other diseases including
NTDs to determine the broad-spectrum activity of TM products.

**Conclusions**

-Are the conclusions supported by the data presented?

-Are the limitations of analysis clearly described?

-Do the authors discuss how these data can be helpful to advance our understanding of
the topic under study?

-Is public health relevance addressed?

Reviewer #2: Yes

Reviewer #3: The discussion and conclusions are well written, the data and results
support the conclusions. The authors have clearly demonstrated how the data could be
helpful in advancing our understanding in traditional medicines and their benefit
for treatment of various diseases including NTDs. made by the authors.

The public health relevance of the study is clearly indicated and the authors have
emphasized the benefit of traditional medicines in Ghana for the treatment of NTDs
and how it could be explored for drug discovery and development for NTDs.

**Editorial and Data Presentation Modifications?**

Reviewer #2: Accept

Reviewer #3: COMMENTS ON REVISED VERSION OF MANUSCRIPT OR RESUBMISSION 

The authors have adequately responded to the comments by all three reviewers and have
appropriately revised the first manuscript to meet my expectations.

I therefore RECOMMEND the revised manuscript to be accepted for publication after all
minor corrections have been made. 

GENERAL CORRECTIONS

1.Pg 4, Line 85 reads … “ ivermectin and albendazole for ONCHO and LF” The sentence
should rather read “ ivermectin and albendazole for LF and ivermectin for ONCHO

2.Pg 21 line 388 column 2, “the” should be deleted

**Summary and General Comments**

Reviewer #2: (No Response)

Reviewer #3: The authors addressed an important and interesting subject that provides
useful evidence for the development of traditional medicines (TMs) for Neglected
Tropical Diseases (NTDs), as well as their potential for policy direction towards
harnessing maximum benefit from TMs in Ghana and globally. However, these resources
are still deployed and practice in indigenous settings, this emphasizes the need for
research and development support for TMs towards the developments of new drugs for
the control and elimination of NTDs.

Extracts of fifteen traditional medicines were investigated for treatment of NTDs
including schistosomiasis, onchocerciasis and lymphatic filaliasis (LF). Some
extracts were found to be efficacious against the causative agents of
schistosomiasis and onchocerciasis and animal African Trypanosomiasis (AAT). The
evaluation of the activity of the extracts showed that some TMs were very active
against some of the disease pathogens while other were not potent. The authors have
demonstrated the need to evaluate for activity of various TMs against other diseases
including NTDs to determine the broad-spectrum activity of TM products.

The findings of this study demonstrates the significant role of traditional medicines
and the prospects towards drug discovery and development of alternate medicine for
NTDs

PLOS authors have the option to publish the peer review history of their article
(what does this mean?). If published, this will
include your full peer review and any attached files.

If you choose “no”, your identity will remain anonymous but your review may still be
made public.

**Do you want your identity to be public for this peer review?** For
information about this choice, including consent withdrawal, please see our
Privacy Policy.

Reviewer #2: Yes: Adela Ngwewondo

Reviewer #3: No
---

## [Author Response · Author response to Decision Letter 1]

17 Oct 2020

letter PNTD-D-20-00655R2_edit.docx
---

## [Editor Report · Decision Letter 2]

26 Oct 2020

Dear Prof Osei-Safo,

We are pleased to inform you that your manuscript 'Antischistosomal, antionchocercal
and antitrypanosomal potentials of some Ghanaian traditional medicines and their
constituents' has been provisionally accepted for publication in PLOS Neglected
Tropical Diseases.

Best regards,

Sabine Specht

Associate Editor

Francesca Tamarozzi

Deputy Editor

---

## [Editor Report · Acceptance letter]

3 Dec 2020

Dear Prof Osei-Safo,

We are delighted to inform you that your manuscript, "Antischistosomal,
antionchocercal and antitrypanosomal potentials of some Ghanaian traditional
medicines and their constituents," has been formally accepted for publication in
PLOS Neglected Tropical Diseases.

Best regards,

Shaden Kamhawi

co-Editor-in-Chief

Paul Brindley

co-Editor-in-Chief
